# A New Method of Obtaining High Purity Nickel(II) Perrhenate from Waste

**Katarzyna Leszczyńska-Sejda \*, Grzegorz Benke, Dorota Kopyto, Joanna Malarz, Mateusz Ciszewski** 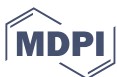 **and Karolina Goc**

Łukasiewicz Research Network—Institute of Non-Ferrous Metals, Sowińskiego 5, 44-100 Gliwice, Poland; grzegorz.benke@imn.lukasiewicz.gov.pl (G.B.); dorota.kopyto@imn.lukasiewicz.gov.pl (D.K.); karolina.goc@imn.lukasiewicz.gov.pl (K.G.)
\* Correspondence: katarzyna.leszczynska-sejda@imn.lukasiewicz.gov.pl

**Abstract:** The article presents a new method of producing anhydrous nickel(II) perrhenate of high purity, entirely from waste from the national Cu industry. This method consists mainly of the reaction of water-washed nickel(II) oxide (obtained by purification in a mixture of alcohols, and subsequent roasting of the Ni-containing sulfate semi-finished products (NSP) at 1200 °C) with perrhenic acid (obtained using the ion exchange method). After the dissolution of nickel(II) oxide in the acid (at a temperature in the range of 60–80 °C) and obtaining a pH of 5–8, the solution is sent to evaporate to dryness, also at a temperature not exceeding 80 °C. The obtained crude nickel(II) perrhenate is washed with methanol and subsequently dried at 160 °C to obtain its anhydrous form, with the following composition: 10.5% of Ni; 66.6% of Re; <5 ppm of Bi, As, Zn and Cu; <10 ppm Co, Mg, Fe, K, Pb, Na, Ca and Mo. Importantly, this composition allows for the use of the compound for the production of superalloys and catalysts. A patent application and a technological scheme were prepared for the developed method. It consists of seven technological operations, including six based on processes in the field of hydrometallurgy, and one in the field of pyrometallurgy (roasting).

**Keywords:** nickel(II) perrhenate; rhenium; nickel; perrhenic acid; superalloys; waste; copper industry





## 1. Introduction

Rhenium is one of the rarest metals in the Earth's crust [1,2]. In a powdered form, it is characterized by a silvery-gray color. It has a hexagonal network (A3), which it retains up to the melting point—3453 K (3180 °C) (only tungsten and carbon have higher melting points), and thanks to which it is not subject to ductile–brittle transformation [3–5]. Rhenium, thanks to its resistance to high temperatures and a very good strength, is used in the production of superalloys, mainly nickel-based ones, from which turbine blades for aircraft engines and gas turbines are made [5–7]. The addition of 1–7% of Re to the nickel-based alloy not only contributes to improving its high temperature strength, but also prevents fatigue cracking [1,3,8]. These superalloys are monocrystalline, have high corrosion resistance, and using them in hot areas of an engine increases the operating temperature, which in turn generates a lower fuel consumption. A similar effect was observed for turbines covered with a coating containing rhenium. As an addition to alloys, rhenium is extremely desirable because it causes a significant increase in plasticity and tensile strength, even after reaching the recrystallization temperature [1,3,8]. Rhenium is also used in catalysts as a component of processes such as: hydrodesulfurization, hydrogenation, hydrocracking, oxidation and reforming. Rhenium catalysts are characterized by high selectivity and extremely high resistance to nitrogen, sulfur and phosphorus [9–11]. There are numerous combinations of rhenium with nickel, cobalt, platinum, palladium and silver used in catalysis [1,9,10].

Rhenium forms stable salts with numerous metals in its seventh oxidation state, including nickel(II) [12–14], which are called perrhenates [15–17]. They are mostly obtained

as a result of the reaction between perrhenic acid and an oxide, hydroxide or carbonate [18–20]. Another possible method is a reaction of $Re_2O_7$ with a metal oxide or an exchange between salts [20–22].

Nickel(II) perrhenates have an orthorhombohedral structure in a tetrahydrate form, and after drying at 160 °C, an anhydrous form is obtained (strongly hygroscopic) which crystallizes into a trigonal lattice at the same time. The dihydrate and tetrahydrate forms of nickel(II) perrhenate are also described in the literature [12]. The literature also describes the ferromagnetic properties of $Ni(ReO_4)_2$, the Curie temperature of 12.5 K, and the solubility and color of all forms of $Ni(ReO_4)_2$. Thus, $Ni(ReO_4)_2 \cdot 4H_2O$ is a green substance with a solubility in water (at 30 °C) of 85.0%. In the case of $Ni(ReO_4)_2 \cdot 2H_2O$ we are dealing with a light green salt with a solubility in water (at 30 °C) of 80.3%. On the other hand, the anhydrous form is a yellow substance with a water solubility (at 30 °C) of 75.6% and a density of 3.95 g/cm$^3$ [7,12].

The first reports on the production of nickel(II) perrhenate in the reaction of perrhenic acid with nickel(II) carbonate or hydroxide come from the 1930s [23–25]. They were described by two groups of scientists, i.e., H.V.A. Briscoe., P.L. Robinson, A.J. Rudge, P. Robinson and A. Rudge in 1931, and E. Wilke-Dorfurt and T. Gunzert in 1933 [23,24].

In 1949, W. Smith and G. Maxwell presented detailed information on nickel(II) perrhenate, indicating that this compound can be obtained by neutralizing perrhenic acid with nickel(II) carbonate or hydroxide, and $Ni(ReO_4)_2 \cdot 4H_2O$ obtained in in this way must be dried at 105 °C to obtain its anhydrous form. This method is sufficient to obtain the salt needed for the conducted research, but it is not suitable for a large-scale production, because it does not solve the problem of waste management, and also uses commercial rhenium and nickel substrates [25].

As it is known from the literature, there is a method of industrial production of anhydrous nickel(II) perrhenate, according to which $Ni(ReO_4)_2$ can be obtained in the reaction of perrhenic acid, containing 15–500 g/dm$^3$ of Re, with commercial nickel compounds, used with a large stoichiometric excess, such as NiO, $Ni(OH)_2$ and $NiCO_3$. The solution obtained in this way, with a minimum pH of 5, is directed to evaporate to dryness, while maintaining a temperature of 80–100 °C and reduced pressure of 0.03–0.05 MPa, in order to separate the hydrated form of nickel(II) perrhenate. It is then sent for drying in the temperature range of 100–160 °C. The described method is characterized by the management of all waste solutions and solid wastes, and allows for the obtainment of anhydrous nickel(II) perrhenate containing a minimum of 10.5% of Ni and 66.5% of Re, with virtually no loss of nickel and rhenium. In this case, the source of nickel are commercial compounds [12,18].

Other reports describe the production of anhydrous nickel(II) perrhenate using ion exchange. According to the description, nickel(II) ions, derived from nickel(II) salts (sulfate and nitrate), are sorbed using a strongly acidic resin and subsequently eluted with perrhenic acid (with a concentration of 400–900 g/dm$^3$ of Re). This method makes it possible to obtain a product with a content of 10.5% of Ni and 66.6% of Re. However, it is time-consuming and consists of quite a few stages. Commercial salts are also the source of nickel in this method. Anhydrous nickel(II) perrhenate is then used to obtain Re-Ni powder, which is a substrate for the production of heavy alloys, e.g., 77W-20Re-3Ni [12,18,19].

The previously described methods use commercial nickel compounds. Due to the shortage of nickel on global markets, as well as the recognition of this metal in 2023 as a critical and strategic material for the EU [26–28], research was undertaken regarding the possibility of producing nickel(II) perrhenate entirely from waste.

## 2. Materials and Methods

### 2.1. Materials

Perrhenic acid was produced from ammonium perrhenate (Metraco KGHM Polska Miedź S.A., Lubin, Poland), obtained from an acid effluent from a Polish copper smelting plant, containing 10–30 mg/dm$^3$ of Re. Rhenium sorption was carried out using a weakly basic ion exchange resin, A170 (Purolite, King of Prussia, PA, USA, hydroxide form), and

the elution was carried out with an aqueous solution of ammonia (25%, Chempur, Piekary Śląskie, Poland, p.a.) [29,30]. Ammonium perrhenate ($NH_4ReO_4$) of catalytic purity was crystallized from the aqueous ammonia solutions, which was subsequently dissolved in water and sorbed ammonium ions, this time using a strongly acidic cation exchange resin, C160 (Purolite, USA, hydrogen form) [31]. The remaining after-sorption solutions were evaporated in order to obtain the desired concentration of rhenium. In this way, perrhenic acid was obtained, containing: 300 g/dm$^3$ of Re, <0.0001% of Ca, <0.0005% of K, <0.0001% of Mg, <0.0001% of Cu, <0.0001% of Na, <0.0001% of Mo, <0.0001% of Ni, <0.0001% of Pb, <0.0001% of Fe, <0.0002% of $NH_4^+$, <0.0001% of Bi, <0.0001% of Zn, <0.0001% of W, <0.0001% of As and <0.0001% of Al.

In the case of nickel, Ni-containing sulfate semi-finished products (NSP) from the Polish Cu industry were selected for the study. In Poland, these semi-finished products are produced during the Cu electrorefining process in all three copper smelters. As part of the research, seven of the above-mentioned materials were selected, which were from the resources of Łukasiewicz-IMN, originating from various places and periods of production.

The following materials were also used in the tests: sulfuric acid (95%, Chempur, Poland, p.a.), nitric acid (65%, Chempur, Poland, p.a.), aqueous solution of hydrogen peroxide (30%, P.P.H. Stanlab, Lublin, Poland, p.a.), demi water (<2 µS/cm; Łukasiewicz-IMN, Gliwice, Poland), methyl orange (85%, Merck, Warszawa, Poland, ACS reagent), sodium hydroxide (Stanlab, Poland, p.a.) and organic solvents (ethanol (99.8%, Chempur, Poland), methanol (99.8%, Chempur, Poland), glycerol (>99.5%, Chempur, Poland), ethylene glycol (99.9%, Chempur, Poland) and acetone (99.6%, Chemland, Stargard, Poland)).

### 2.2. Apparatus

A pilot plant belonging to Łukasiewicz-IMN (Gliwice, Poland), as shown in Figure 1, was used in the research on the production of perrhenic acid.

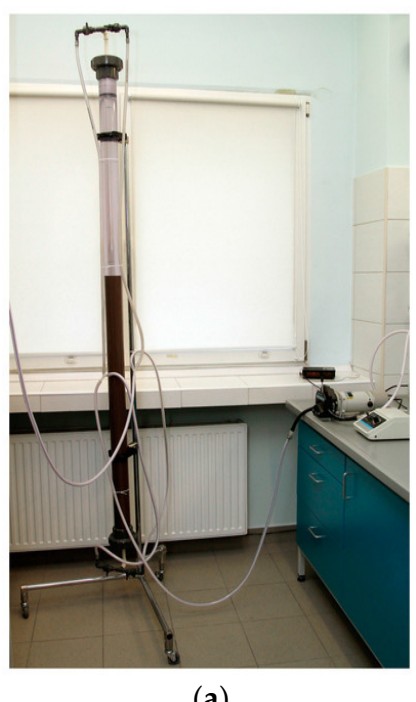
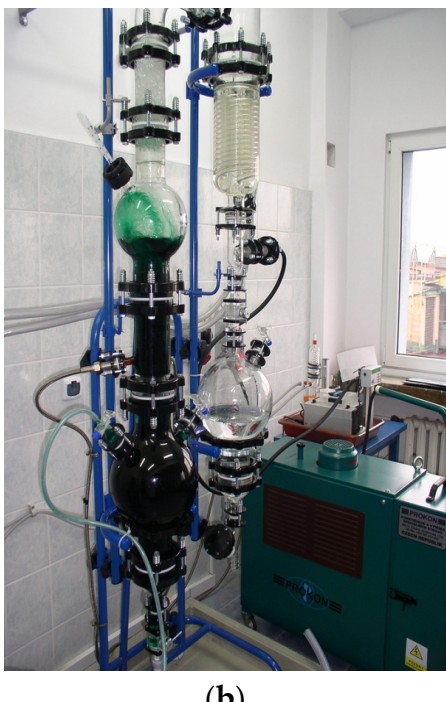

| (a) | (b) |

**Figure 1.** Main elements of the installation for the production of perrhenic acid are: (**a**) an ammonium ion sorption column; (**b**) a vacuum evaporator.

Drying tests were performed using a WPS 210 S moisture analyzer (Mettler Toledo, Poznań, Poland). The roasting tests were carried out using a furnace in a nitrogen atmosphere (Carbolite HST 1200, Sheffield, UK).

### 2.3. Analytical Methods

All analyses of nickel(II) perrhenate, NSP, perrhenic acid and other solutions were performed at the Łukasiewicz Research Network-Institute of Non-Ferrous Metals, Centre of Analytical Chemistry (Gliwice, Poland). The rhenium content in nickel(II) perrhenate, NSP and perrhenic acid were analysed using thin layer X-ray fluorescence spectrometry with the use of a fluorescent X-ray spectrometer (ZSX Primus, Rigaku, Tokyo, Japan). Nickel content in nickel(II) perrhenate was determined using the FAAS (Flame Atomic Absorption Spectroscopy) method. Concentrations of some pollutants (such as As, Cu, Mg, Zn, Ca, Fe, Mo, Pb, Na, Bi, K, Co and K) were determined using the following instrumental techniques: GFAAS (graphite furnace atomic absorption spectroscopy with graphite cells, Z-2000, HITACHI, Tokyo, Japan), ICP-OES (Inductively Coupled Plasma—Optical Emission Spectrometers; ULTIMA 2, HORIBA Jobin-Ivon, Kyoto, Japan) and ICP MS (Inductively Coupled Plasma—Optical Emission Spectrometers; Plasma-Mass Spectroscopy; Nexion, PerkinElmer, Waltham, MA, USA). Rhenium and nickel concentrations in the solutions were determined using the FAAS method, with the use of a THERMO atomic absorption spectrometer SOLAAR S4 (Thermo Fisher Scientific, Waltham, MA, USA), equipped with a flame module and deuterium background correction. Ammonium ions, e.g., in the aqueous solution of perrhenic acid, were determined by a distillation method with a titration after the distillation—the Nessler method. In order to determine the main pollutants, the emission atomic spectrometry with excitation inductively coupled plasma ICP-OES with a Horizon ARL was used.

XRD analyses were performed at the Łukasiewicz Research Network-Institute of Non-Ferrous Metals, Centre of Functional Materials (Gliwice, Poland), based on the interpretation of diffraction patterns prepared with a XRD diffractometer Rigaku MiniFlex 600, equipped with an X-ray tube wavelength of 1.5406 Å, D/TeX silicon strip detector and 2.5″ high-resolution Soller slits on the primary and diffuse beam.

10 g portions of each material was then mixed with distilled water at 90 °C for 10 min. The suspension was filtered. In the filtrate, the amount of acid was determined by titration, using methyl orange as an indicator and a standard solution of NaOH with a concentration of 0.1 mol/dm$^3$.

## 3. Results

### 3.1. Analysis of the Composition of NSP

The selected NSP were subjected to a quantitative analysis to check the content of, above all, nickel, but also selected impurities, such as: Co, Cu, Zn, Mg, Pb, Na, Ca and Fe (Table 1). Because of their form, the acid content was also determined (Table 2).

**Table 1.** Chemical composition of the materials (NSP).

| No. | Composition, % | | | | | | | | |
|-----|------|-------|--------|-------|-------|--------|--------|------|--------|
|     | Ni   | Co    | Cu     | Zn    | Mg    | Pb     | Na     | Ca   | Fe     |
| NSP1 | 27.0 | 0.410 | 0.71   | 0.120 | 0.068 | 0.0070 | 0.0071 | 0.48 | 0.45   |
| NSP2 | 29.8 | 0.510 | 0.58   | 0.140 | 0.074 | 0.0090 | 0.0092 | 1.18 | 0.49   |
| NSP3 | 16.0 | 0.210 | 12.0   | 0.059 | 0.036 | 0.0330 | 0.0095 | 1.58 | 0.23   |
| NSP4 | 23.0 | 0.250 | 1.15   | 0.150 | 0.078 | 0.0180 | 0.0100 | 1.17 | 0.38   |
| NSP5 | 26.5 | 0.450 | 0.62   | 0.120 | 0.068 | 0.0097 | 0.0060 | 0.67 | 0.41   |
| NSP6 | 26.6 | 1.320 | 1.35   | 0.230 | 0.071 | 0.0095 | 0.0450 | 0.69 | 0.51   |
| NSP7 | 22.3 | 0.019 | <0.005 | 0.003 | 0.039 | <0.0005 | 0.1800 | 0.07 | <0.0005 |

After the analysis of the obtained results, it was concluded that these materials are semi-finished products of various composition, which have a good potential for Ni recovery, but are not directly suitable for the production of Ni(ReO$_4$)$_2$. The Ni content ranged from 16 to 30%, while the Co content ranged from 0.019 to 1.32% and Cu from <0.005 to 12%. In the cases of Mg and Pb, the content of these metals did not exceed 0.08% each. On the

other hand, the Na content was <0.2%. The amount of Fe, Ca and Zn was very diverse and amounted to <0.0005–0.51%, 0.07–1.58% and 0.0033–0.23%, respectively. The acid content in all the tested samples varied and ranged from 0.3 to 8.2%. The smallest amount of impurities was found in NSP7, and the highest in NSP3, which contained 12% of Cu. A relatively high cobalt content (1.32%) was determined for NSP6. NSP1, NSP3, NSP4 and NSP6 had a high acid content, and NSP2 and NSP7 materials had a low acid content.

Drying tests were performed for each material (NSP1-NSP7) using batches of 10 g; the drying temperature range was 40–200 °C. Figure 2 shows four selected diagrams of NSP materials (NSP1, NSP3, NSP5, NSP7).

For all materials, a significant, though highly differentiated, mass loss was observed. For NSP1 and NSP6, the mass loss was 6%, and for NSP2 it was the lowest, i.e., ~2%. In the cases of NSP3, NSP4, NSP5 and NSP7, mass losses were high and amounted to 20, 12, 20 and ~35%, respectively.

**Table 2.** Acid content in NSP.

| No. | Sample Weight, g | Acid Weight, g | Acid Content in the Sample, % |
|---|---|---|---|
| NSP1 | 10.28 | 0.59 | 5.7 |
| NSP2 | 9.98 | 0.03 | 0.3 |
| NSP3 | 10.11 | 0.83 | 8.2 |
| NSP4 | 10.12 | 0.74 | 7.4 |
| NSP5 | 10.33 | 0.33 | 3.2 |
| NSP6 | 10.04 | 0.42 | 4.1 |
| NSP7 | 10.05 | 0.02 | 0.2 |

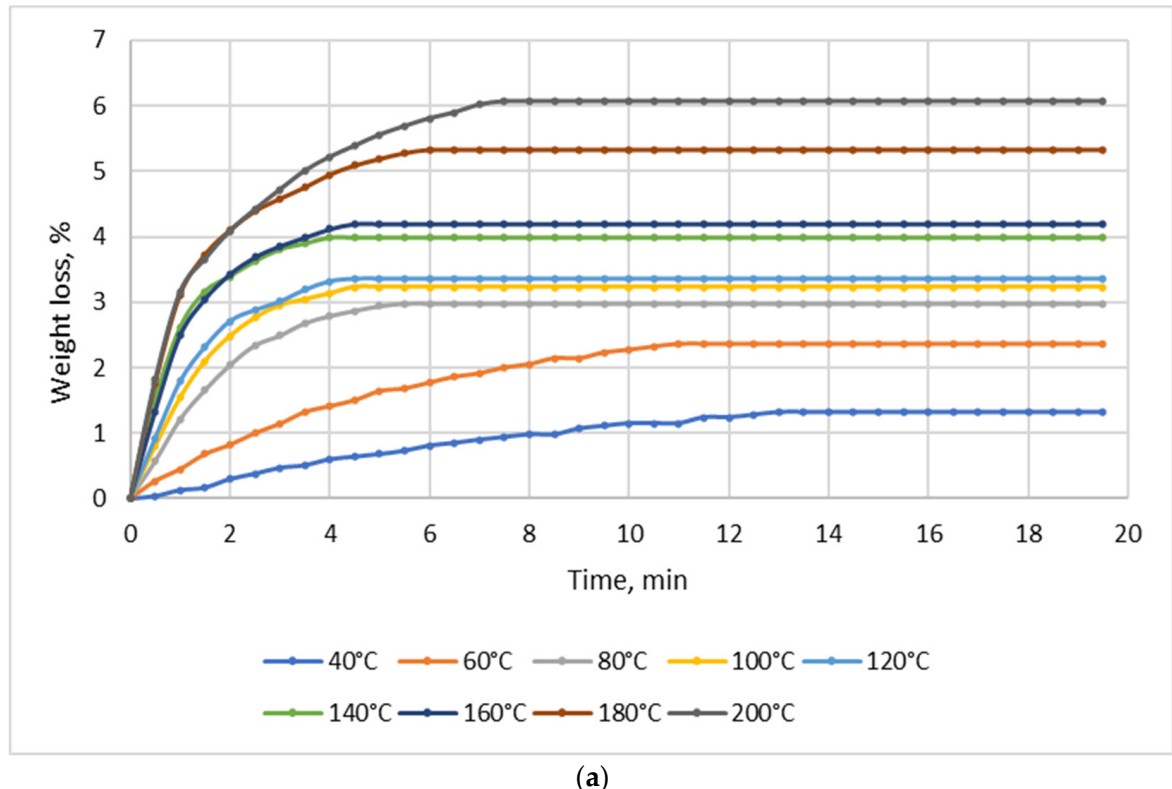

(**a**)

**Figure 2.** *Cont.*

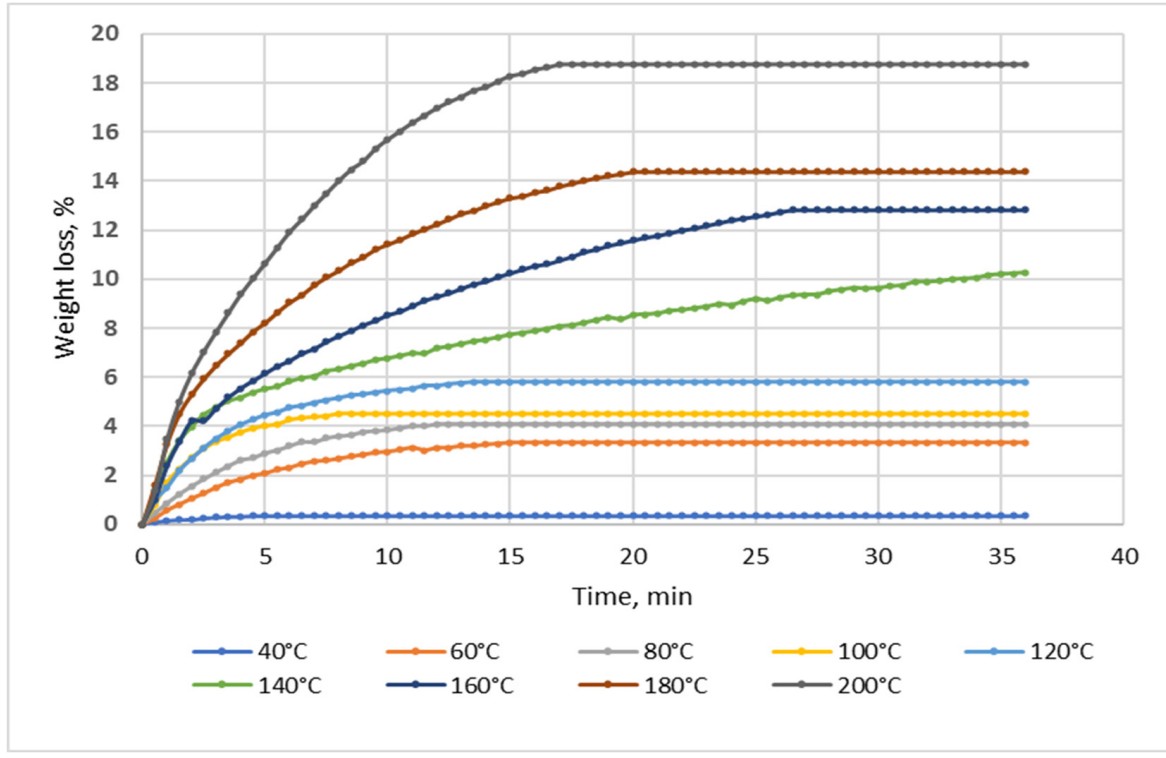

(**b**)

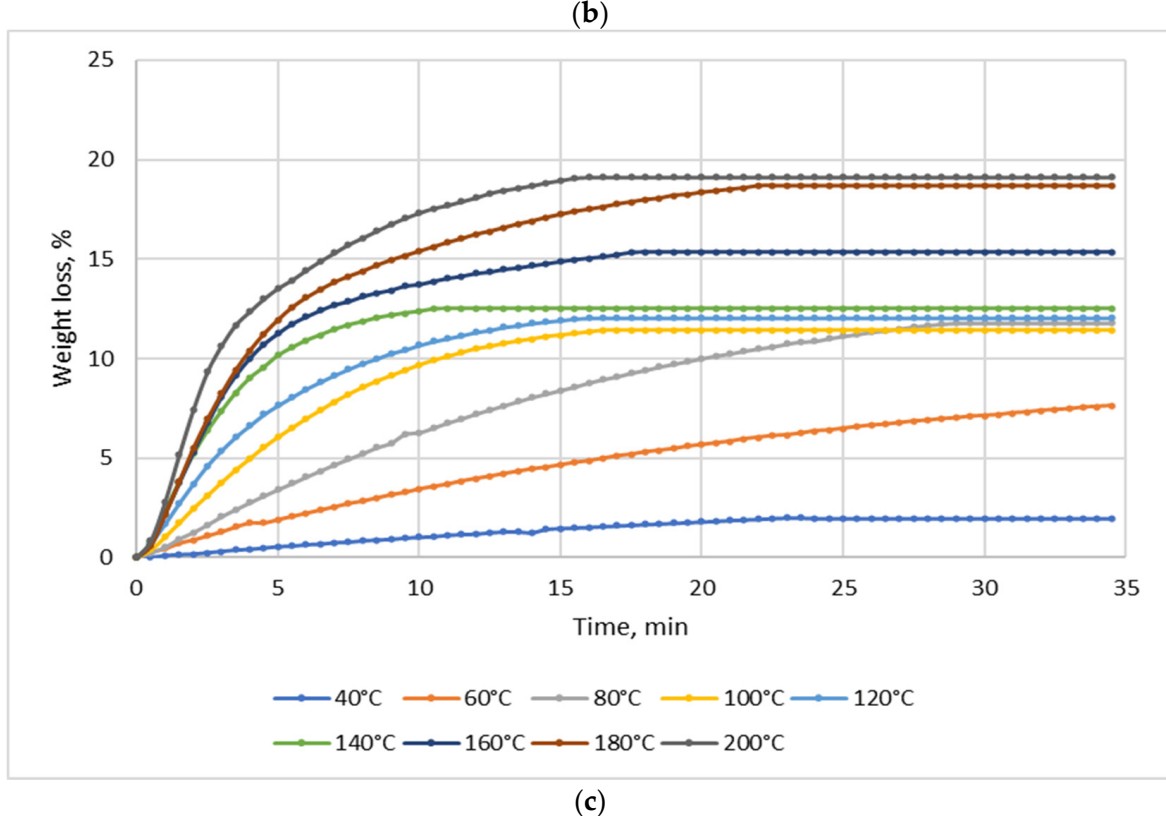

(**c**)

**Figure 2.** *Cont.*

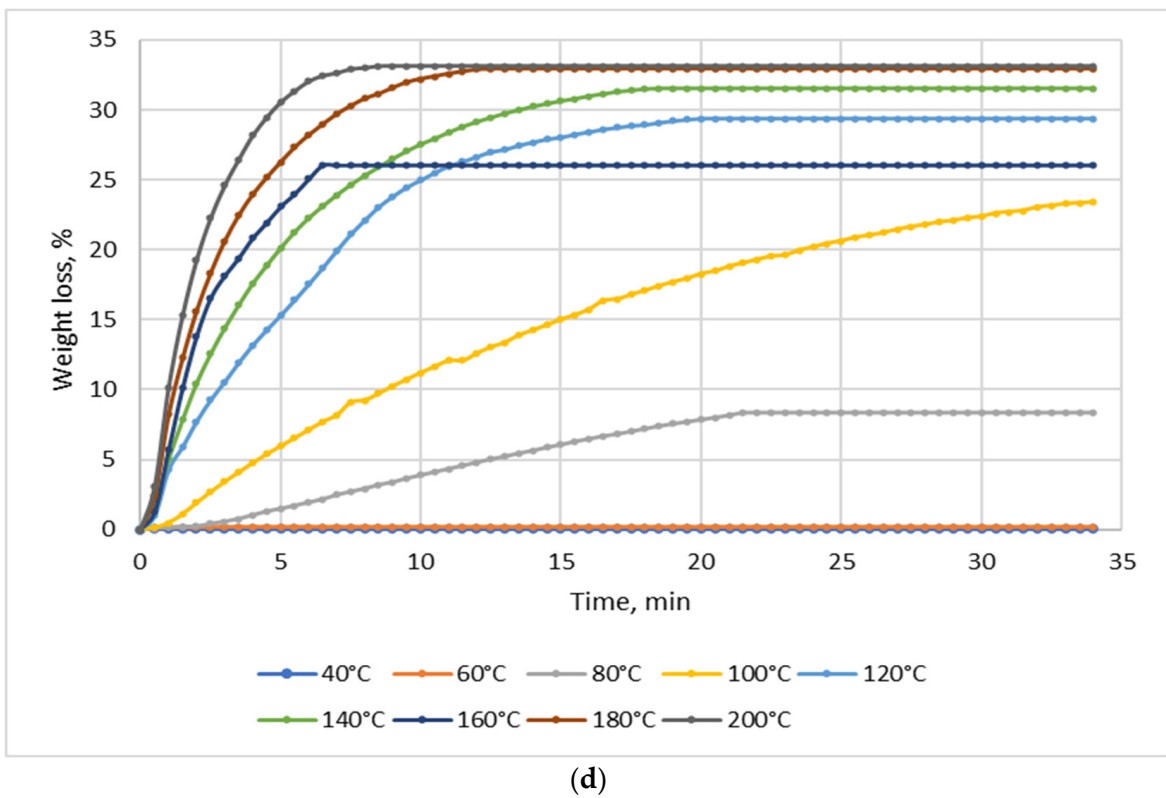

(**d**)

**Figure 2.** Dependence of the effect of drying temperature on NSP' mass losses: (**a**) NSP1; (**b**) NSP3; (**c**) NSP5; (**d**) NSP7.

These materials were subjected to XRD analysis. Figure 3 shows patterns for three NSP materials (NSP2, NSP3, NSP5) dried at 200 °C and for one NSP material (NSP7) dried at 100 °C. These X-ray phase diagrams provided a firm basis for the statement that all the tested substances are materials of different composition, which, however, have common components. All graphs show the presence of nickel(II) sulfate in the forms of various hydrates, mainly $NiSO_4 \cdot H_2O$. In addition, the phase diagrams of NSP1, NSP2, NSP7 show the oxide and sulfide phases of nickel, and in the case of NSP3, NSP4, NSP6, only the sulfide phase bound with nickel. However, for NSP7, only $NiSO_4 \cdot 6H_2O$ is found.

As can be seen, materials with such a composition are not directly suitable for the production of high-purity $Ni(ReO_4)_2$ and require appropriate preparation. All materials were sent for further research.

*3.2. Purification of NSP*

NSP samples in their original form, i.e., without drying, were subjected to purification studies. The tests were carried out at room temperature, using 5 g of the material sample and 25 cm$^3$ of the selected solvent. The sample with the solvent was mixed for 15 min, then filtered, dried at room temperature, and the resulting solution was analyzed for the content of selected impurities, i.e., cobalt, copper, sodium, calcium, iron and nickel. For all materials and impurities, the purification efficiency was calculated, which was defined as: the ratio of the mass of metal in the solvent solution to the mass of metal in the material sent for purification. The test results are shown in Figure 4. In the tests using methanol and ethanol, the concentrations of cobalt, copper, sodium, calcium and iron were analyzed in the solution after purification. On the other hand, in the solutions formed after purification with the use of glycerol and ethylene glycol, the concentrations of sodium, calcium and iron were determined.

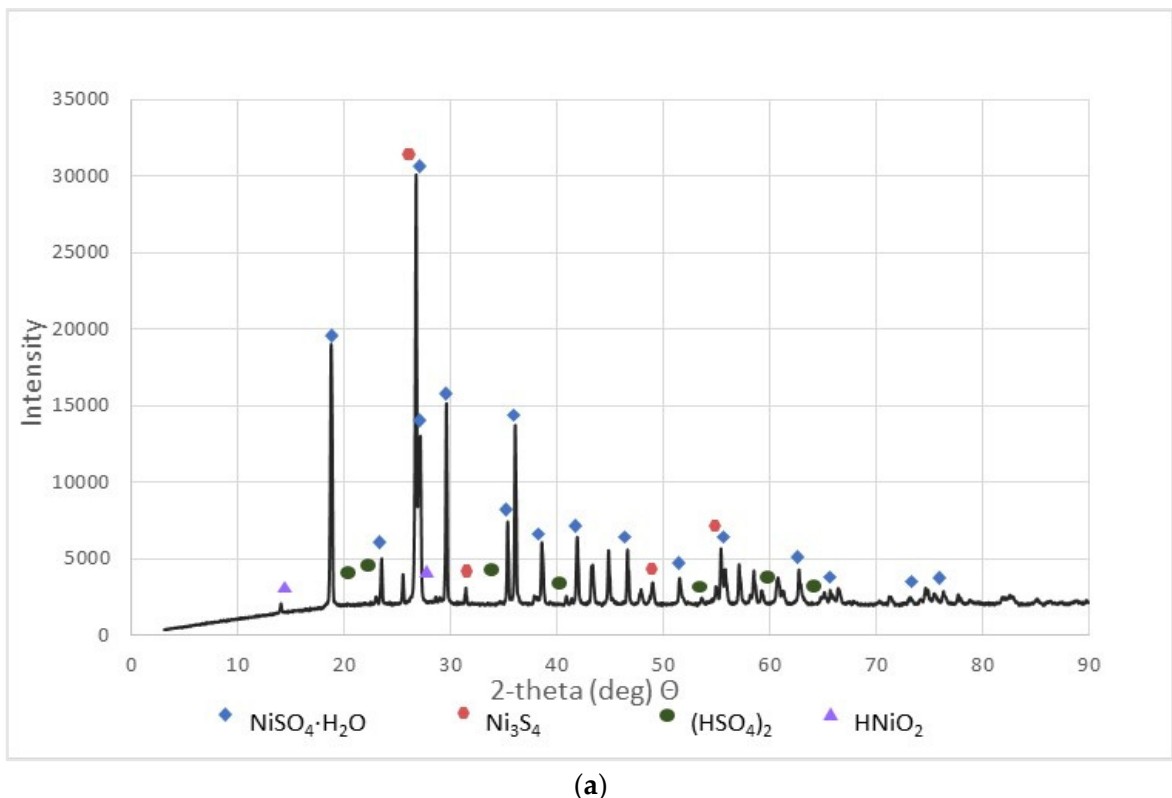

(**a**)

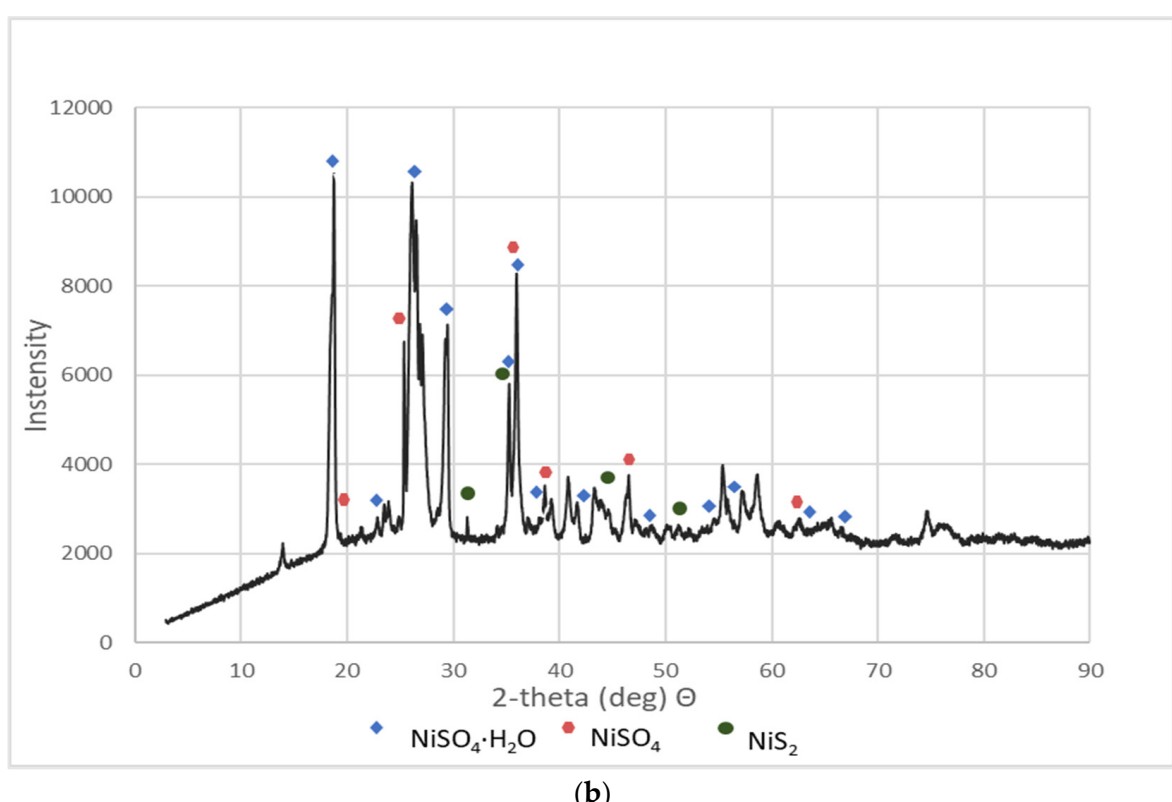

(**b**)

**Figure 3.** *Cont.*

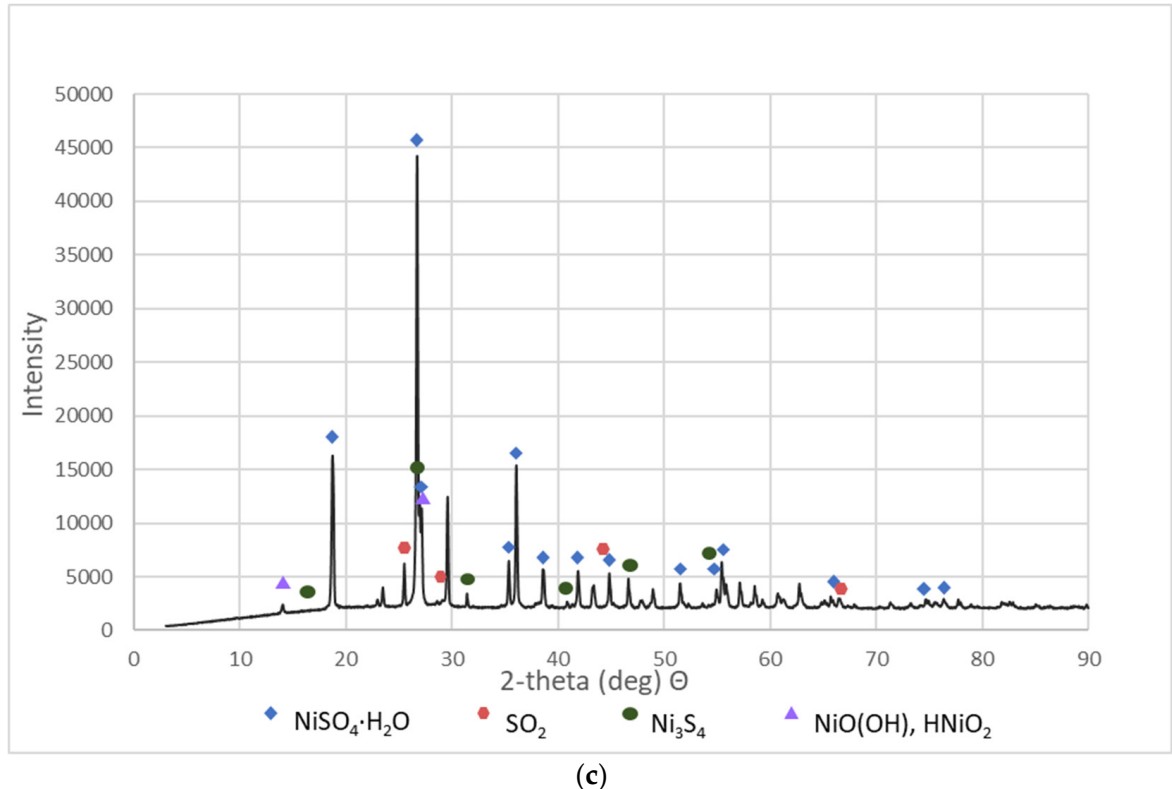

(**c**)

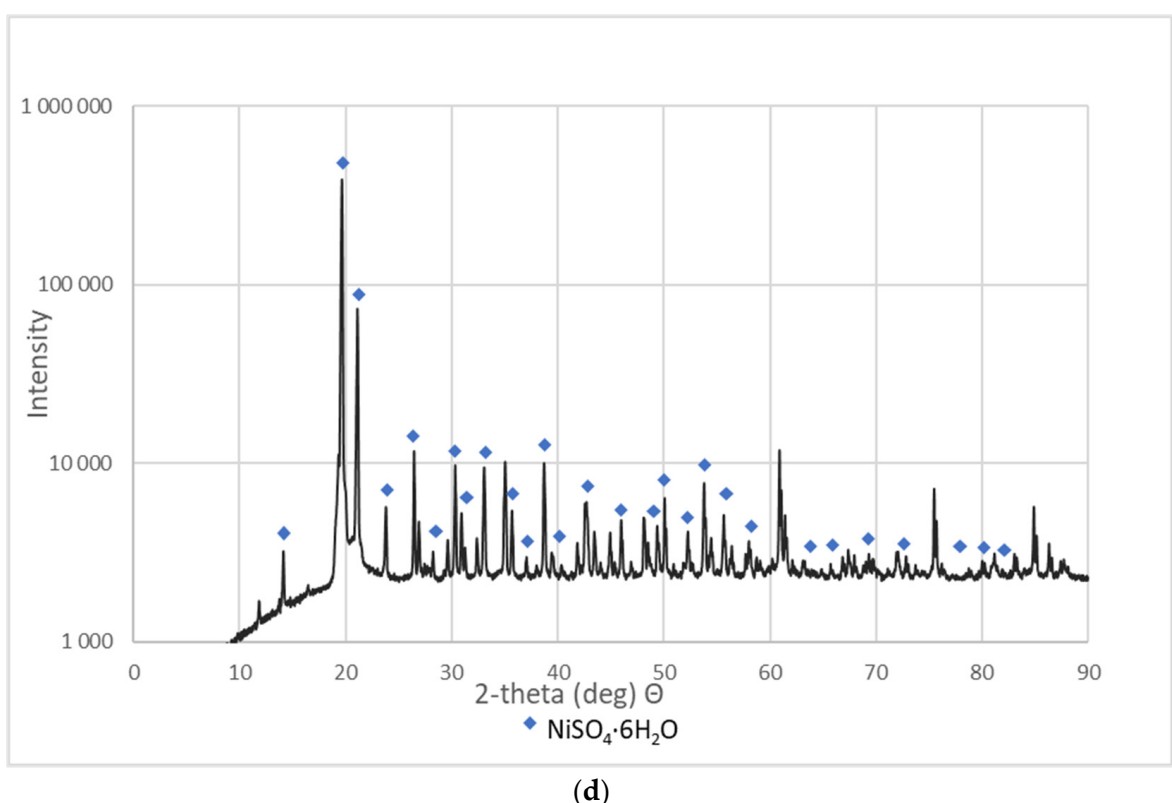

(**d**)

**Figure 3.** XRD patterns of the selected NSP: (**a**) NSP2; (**b**) NSP3; (**c**) NSP5; (**d**) NSP7.

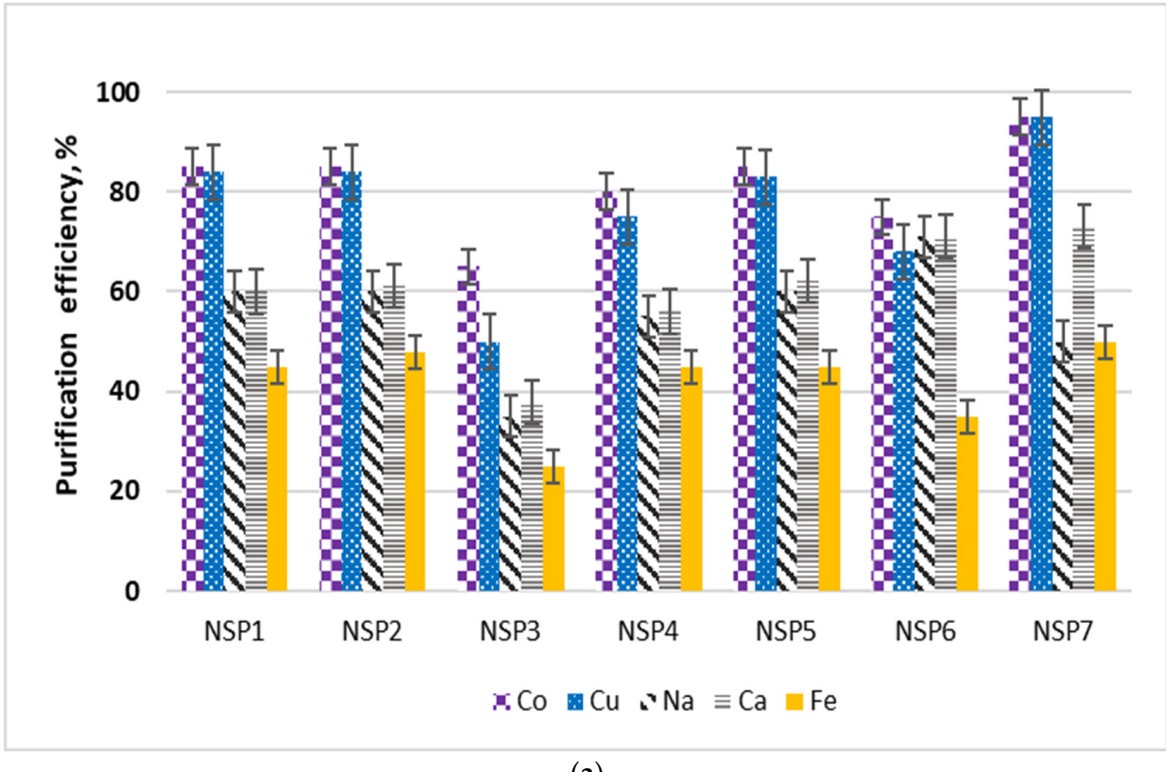

(**a**)

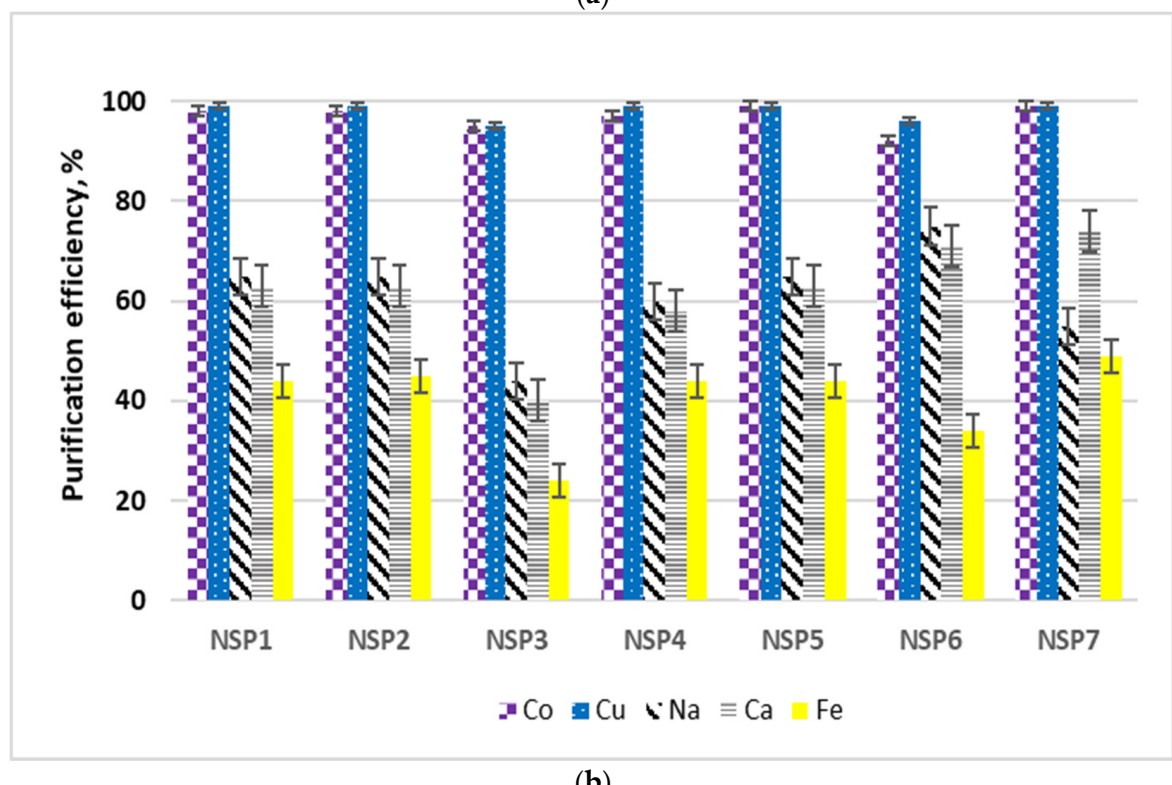

(**b**)

**Figure 4.** *Cont.*

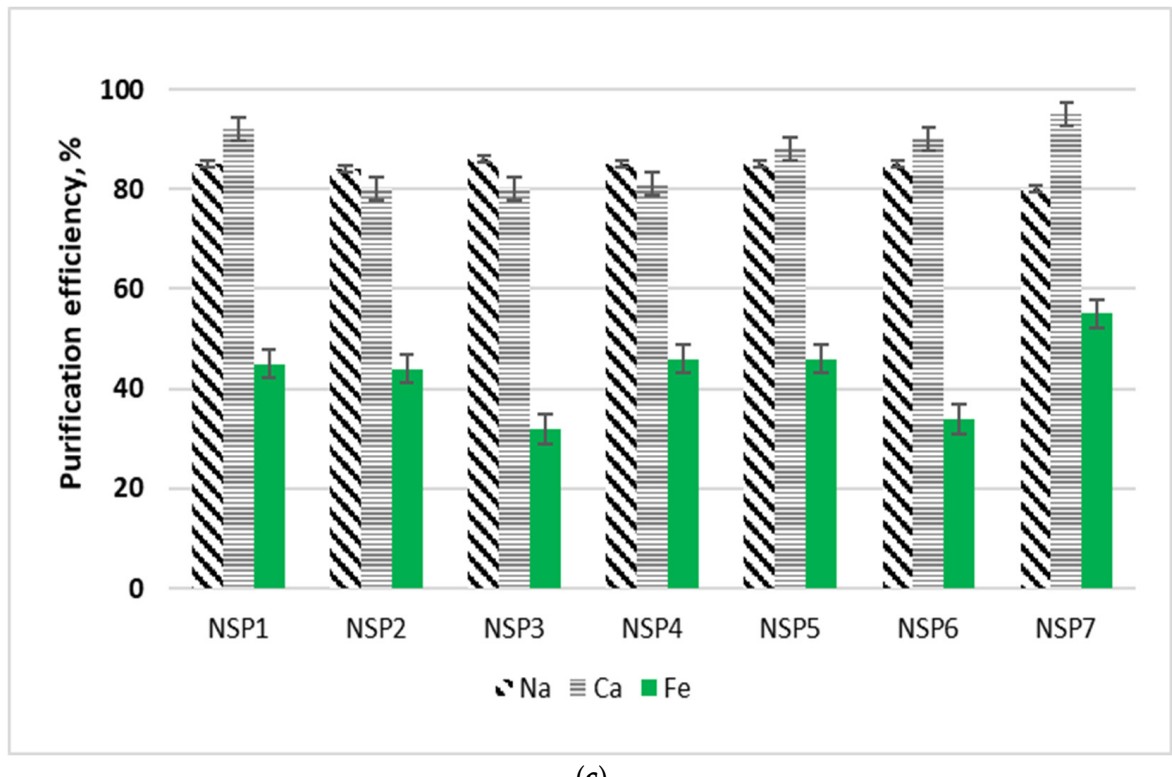

(**c**)

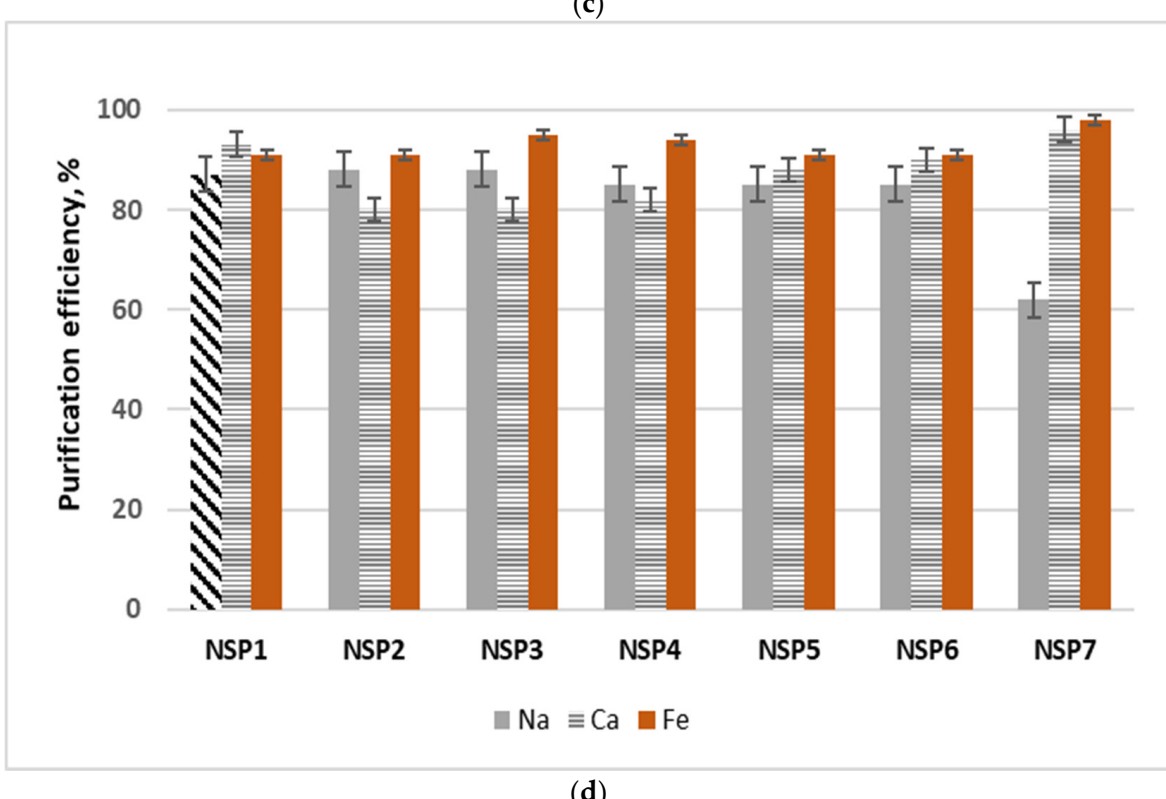

(**d**)

**Figure 4.** Diagrams of purification of NSP in various organic solvents: (**a**) ethanol; (**b**) methanol; (**c**) glycerol; and (**d**) ethylene glycol.

Nickel was analyzed in all solutions; at this stage, for the selected solvents, nickel losses amounted to 2–6%. The lowest nickel loss was identified while using glycerol. The highest purification efficiency from copper and cobalt (>95% and >90%, respectively) was obtained when methanol was used for the purification. The purification efficiencies from

copper and cobalt while using ethanol were slightly lower than the those obtained with methanol; however, the sodium and calcium purification efficiencies were on average 5% higher. The iron purification efficiency using ethanol or methanol was around 41%. In the case of copper and cobalt, methanol and ethanol turned out to be effective purifying agents; in the case of sodium and calcium it was glycerol; and in the case of iron it was ethylene glycol. Due to the variety of materials, it was considered that, in general, the purification of NSP materials should be carried out with the use of a mixture of alcohols, where a minimum of 5 cm$^3$ of solution should be used for each 1 g of the material, in the following proportions:

- if Co and/or Cu content in the NSP is $\geq$1.0%, a mixture containing $\leq$50% *v/v* of ethanol and $\geq$50% *v/v* of methanol should be used for purification;
- if Co and/or Cu content in the NSP does not exceed 1.0% in total, a mixture containing $\geq$50% *v/v* of ethanol and $\leq$50% *v/v* of methanol should be used for purification;
- if Ca and Na content in the NSP exceed 0.5% in total, a mixture containing $\leq$40% *v/v* of ethanol and $\leq$50% *v/v* of methanol, with the addition of up to 10% *v/v* of glycerol, should be used for purification;
- if Fe content in the NSP exceeds 0.5%, a mixture containing $\leq$40% *v/v* of ethanol and $\leq$50% *v/v* of methanol, with the addition of up to 10% *v/v* of ethylene glycol, should be used for purification;
- if Ca and Na content in the NSP exceeds 0.5% in total, and at the same time the content of Co and Cu exceed 1.0% and the Fe content exceeds 0.5%, a mixture containing $\leq$40% *v/v* of ethanol and $\leq$50% *v/v* of methanol, with the addition of up to 10% *v/v* of glycerol and ethylene glycol, should be used for purification.

For the roasting tests, a 100 g sample of NSP7 was prepared, which was purified using 0.5 dm$^3$ of a mixture of alcohols containing 60% *v/v* of ethanol and 35% *v/v* of methanol, with the addition of 5% *v/v* of glycerol.

### 3.3. Roasting of NSP7

Purified NSP7 was sent for roasting tests using the furnace in a nitrogen atmosphere, using a gas flow of 5 dm$^3$/min, in a temperature range of 600–1200 °C, for an hour. The tests were carried out using a 5 g sample of the material. The obtained roasting products were analyzed for the content of nickel, cobalt, copper, zinc, iron, magnesium, sodium, lead and calcium (Table 3). XRD patterns are shown in Figure 5. Figure 6 presents the appearance of the powders after roasting.

**Table 3.** Composition of the powders after roasting.

| No. | Composition, ppm or % * | | | | | | | | |
|---|---|---|---|---|---|---|---|---|---|
| | Ni * | Co | Cu | Zn | Mg | Pb | Na | Ca | Fe |
| NSP7—600 °C | 38.98 | 10 | | | | | | 10 | |
| NSP7—800 °C | 52.51 | 10 | <1 | <1 | <1 | <1 | <10 | 10 | <1 |
| NSP7—1000 °C | 77.47 | 6 | | | | | | 7 | |
| NSP7—1200 °C | 78.57 | 4 | | | | | | 4 | |

Stoichiometrically calculating from 5 g of nickel(II) sulfate hexahydrate, 1.43 g of nickel(II) oxide should be obtained. Analyzing the results, the same amount was obtained after roasting at 1200 °C. The obtained material was green in color (Figure 6). By roasting at 600 °C, anhydrous nickel(II) sulfate was produced, with a yellow-green color (Figure 6). In the case of the powder obtained at 800 °C, there were spots of green particles, but most of them were black—both colors are attributed to nickel(II) oxide. The material obtained as the result of roasting at 1000 °C indicates obtaining the target oxide, both in color (dark green) and mass (1.45 g). Quantitative analysis (Table 3) and qualitative X-ray phase analysis were performed for each powder. Both analytical techniques confirmed obtaining pure nickel(II) oxide, both during roasting at 1000 and 1200 °C. Summarizing the results of the roasting tests, it can be stated that NiO of stoichiometric composition and high purity

can be obtained using the roasting temperature of 1200 °C, for 1 h, with the gas flow of 5 dm$^3$/min. Using this procedure, successive batches of nickel(II) oxide were produced for the studies on the preparation of nickel(II) perrhenate.

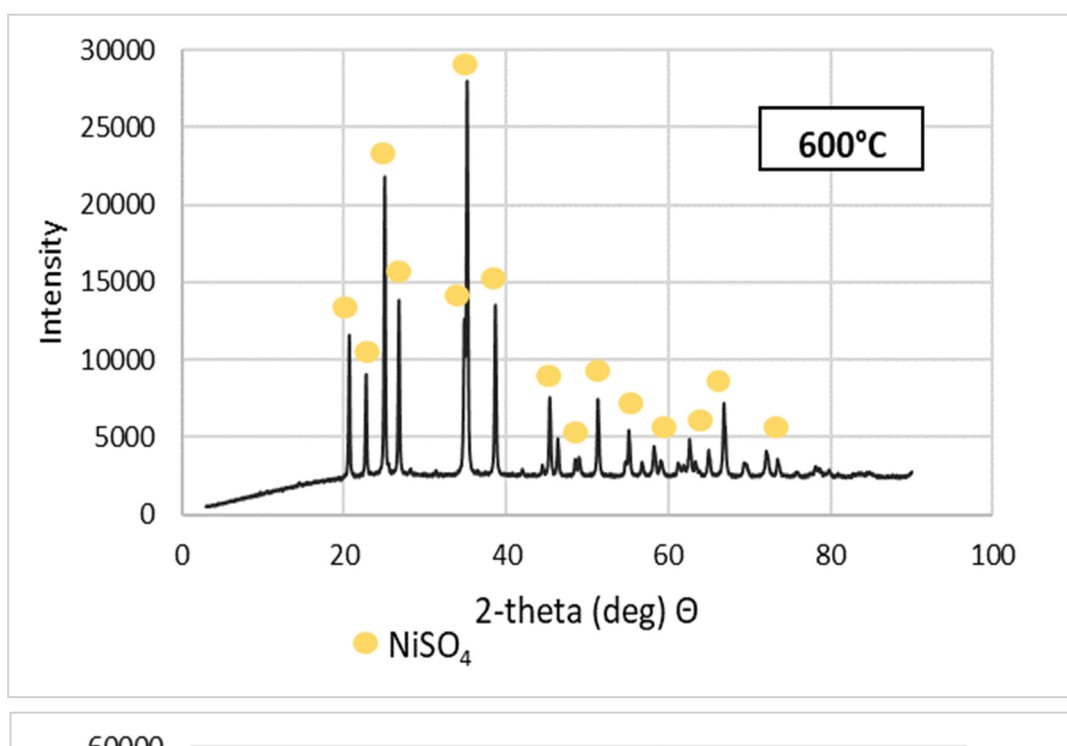

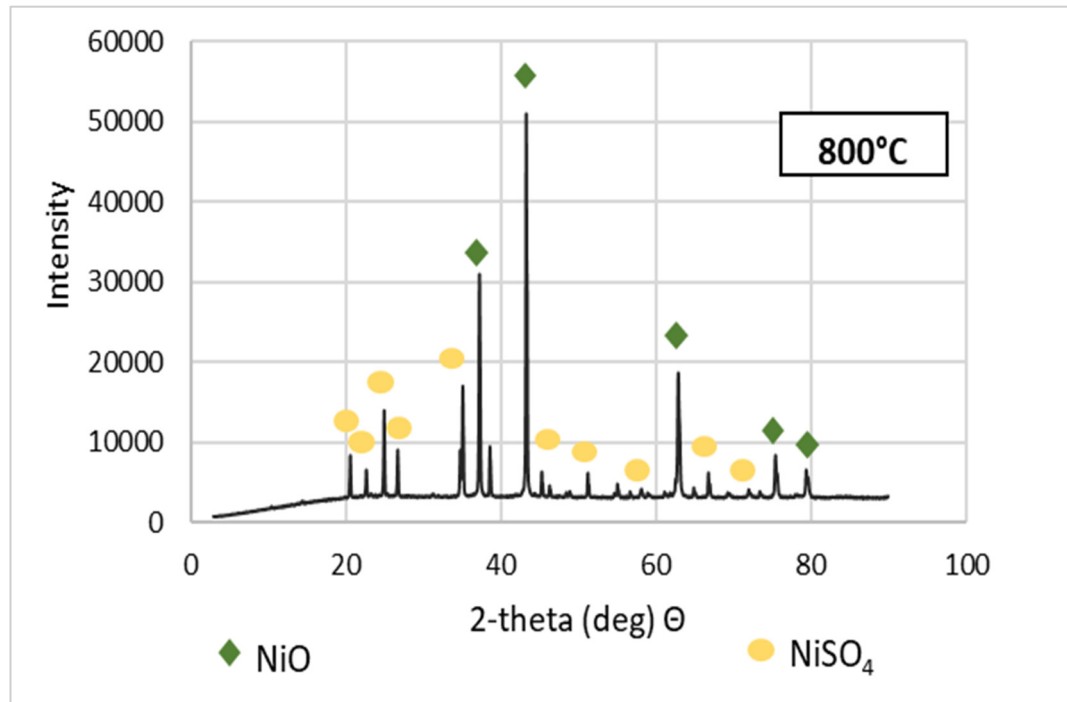

**Figure 5.** *Cont.*

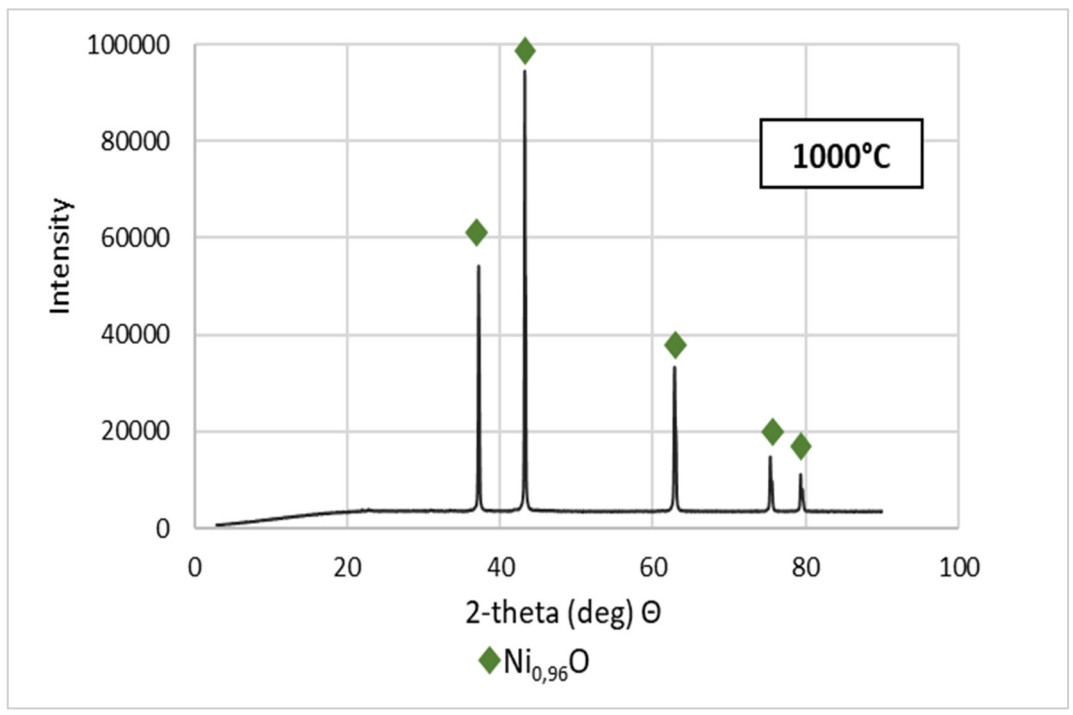

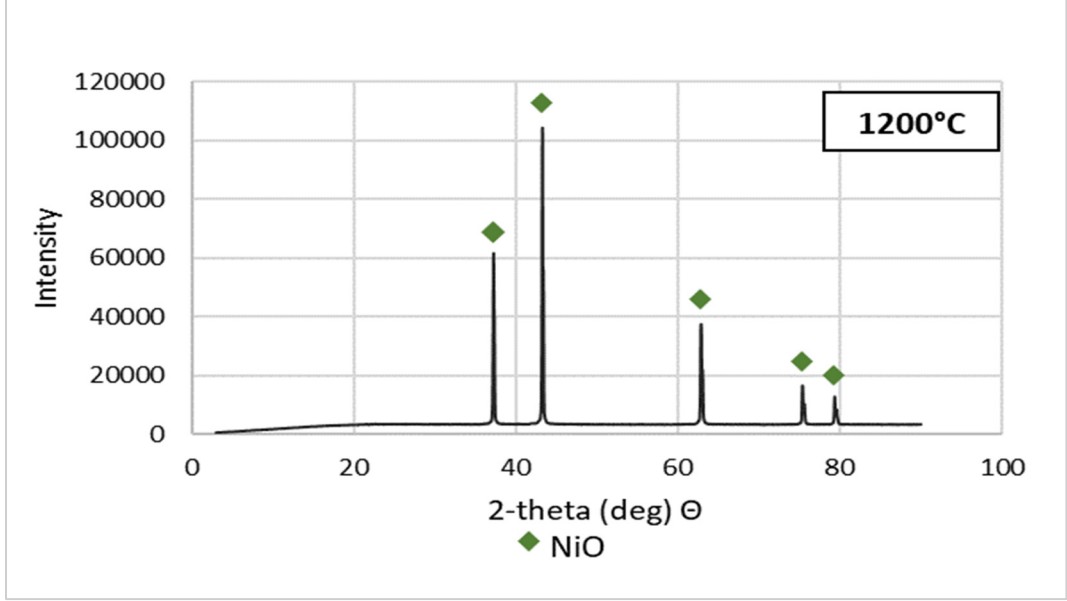

**Figure 5.** XRD patterns of NSP7 after roasting, at different temperature values.

*3.4. Preparation of Nickel(II) Perrhenate*

The tests were carried out using the powder of nickel(II) oxide obtained by roasting at 1200 °C. To the perrhenic acid solution (with the following composition: 300 g/dm$^3$ of Re, <0.0001% of Ca, <0.0005% of K, <0.0001% of Mg, <0.0001% of Cu, <0.0001% of Na, <0.0001% of Mo, <0.0001 % of Ni, <0.0001% of Pb, <0.0001% of Fe, <0.0002% of NH$_4^+$, <0.0001% of Bi, <0.0001% of Zn, <0.0001% of W, <0.0001% of As and <0.0001% of Al) nickel(II) oxide, previously washed with water, was added in portions. The reaction was carried out for 2 h, at a temperature ranging from 60 to 80 °C, until complete dissolution of NiO and at the same time until the acid was neutralized (pH 5–8).

$$2HReO_4 + NiO \rightarrow Ni(ReO_4)_2 + H_2O$$

The resulting solution was evaporated to dryness, also at a temperature not exceeding 80 °C. The precipitate of hydrated, crude nickel(II) perrhenate obtained in this way was divided into 1 g portions and successively washed on a filter with various organic solvents (methanol, ethanol, acetone) in the amount of 10 cm$^3$, thus obtaining finished products. The powders were successively dried at 160 °C and analyzed for the content of the main components, i.e., Ni and Re, and selected impurities: Bi, As, Zn, Cu, Co, Mg, Fe, K, Pb, Na, Ca and Mo. Figure 7 shows a photo of the final product.

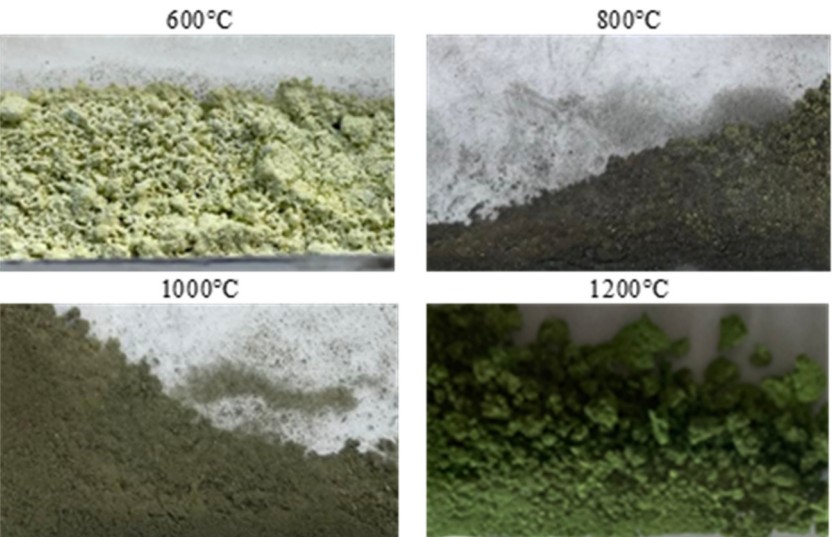

**Figure 6.** Appearance of the powders after roasting.

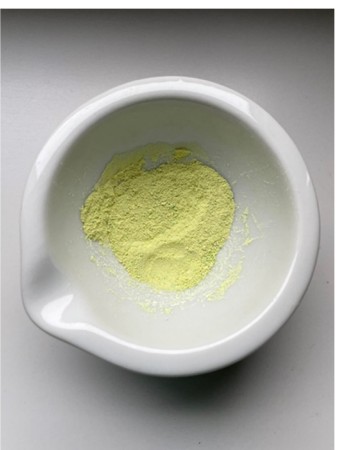

**Figure 7.** Appearance of the final nickel(II) perrhenate.

In this way, anhydrous Ni(ReO$_4$)$_2$ of high purity (suitable for use in the production of superalloys and catalysis) was obtained, with the following composition: 10.5% of Ni; 66.6% of Re; <5 ppm of the following: Bi, As, Zn and Cu; and <10 ppm of the following: Co, Mg, Fe, K, Pb, Na, Ca and Mo. It was also noted that methanol turned out to be the best of the selected solvents. With the use of it, material losses resulting from dissolution were below 4%.

## 4. Conclusions

Using the aforementioned conditions, the new, hybrid technology for obtaining anhydrous nickel(II) perrhenate of high purity, entirely from waste, was created. The source of nickel was Ni-containing sulfate semi-finished products (NSP); a material with a very diverse composition, obtained during Cu production in Poland. In the course of the pre-

sented research, the method of its transformation into the form suitable for the production of nickel(II) perrhenate, i.e., nickel(II) oxide, was developed. Obtaining the appropriate form and purity of nickel(II) oxide was possible through three successive technological operations, i.e., washing of the NSP with the mixture of alcohols (ethanol, methanol and/or glycerol and/or ethylene glycol) (1), roasting of the NSP at the temperature of 1200 °C (2), and washing of the obtained product with water (3). The nickel(II) oxide obtained in this way was the suitable material for the reaction with perrhenic acid. In this way, the Ni-Re solution (4) was formed, which was subsequently evaporated to dryness (5). The obtained precipitate was washed with methanol (6) and then dried (7). The method was developed in accordance with the principles of sustainable development (the 5R's—Refuse, Reduce, Reuse, Repurpose, Recycle), as it manages all waste and thus minimizes the losses of valuable metals, such as rhenium and nickel, to zero.

The developed method was created on the basis of the following assumptions:

- maximizing the utilization of all waste solutions,
- minimizing the losses of valuable components (rhenium and nickel),
- maximizing the use of waste,
- minimizing energy consumption.

Figure 8 presents a diagram of the developed method.

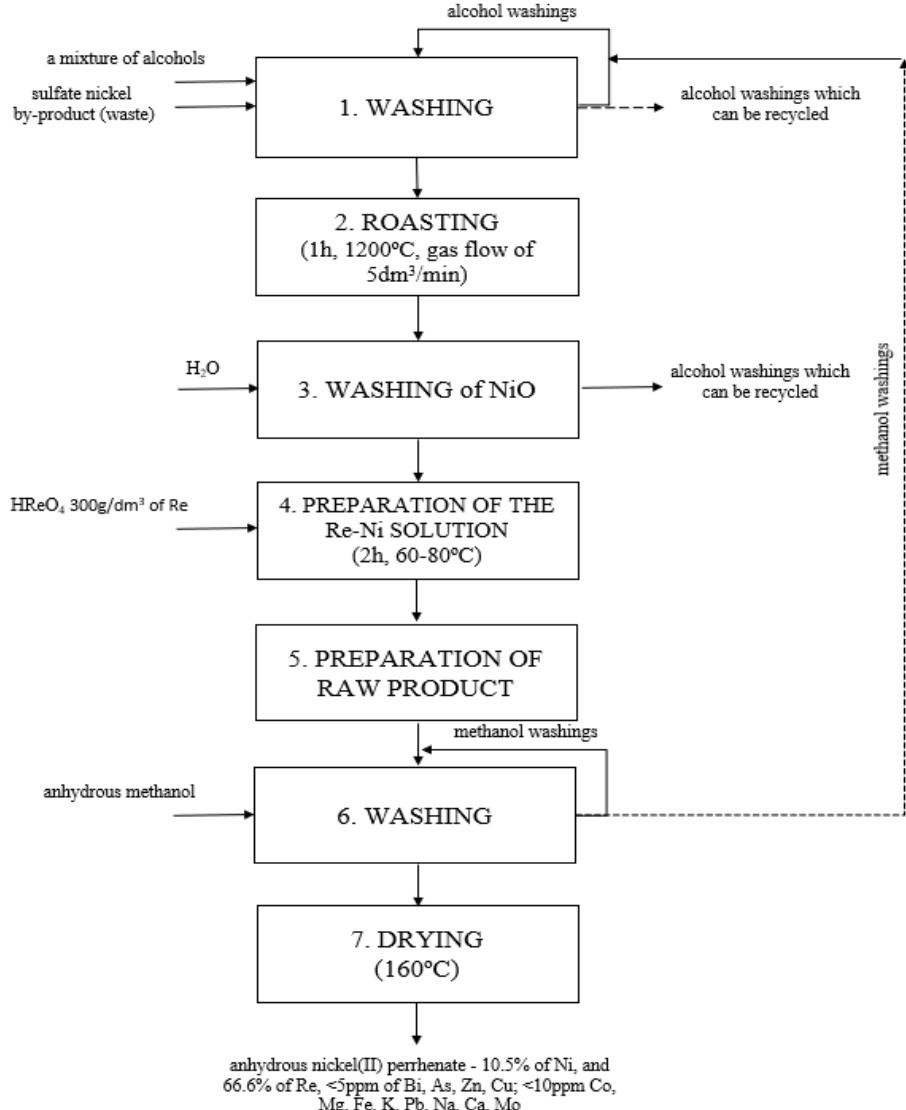

**Figure 8.** Technological scheme of the new hybrid method of obtaining anhydrous nickel(II) perrhenate with high purity from waste.

## 5. Patent

Part of the results of the work presented in this publication is the material submitted for patenting in the Patent Office of the Republic of Poland, under the number P.444211, on March 23, 2023, entitled: Sposób wytwarzania wysokiej czystości bezwodnego renianu(VII) niklu(II) z odpadów i półproduktów (English title: A method of producing high purity anhydrous nickel(II) perrhenate from waste and semi-finished products).

**Author Contributions:** Conceptualization, K.L.-S., M.C. and G.B.; methodology, K.L.-S.; validation, K.L.-S., D.K., M.C. and J.M.; investigation, K.L.-S., J.M. and M.C.; resources, K.L.-S., G.B. and D.K.; data curation, K.L.-S., G.B. and D.K.; writing—original draft preparation, K.L.-S.; writing—review and editing, K.L.-S., K.G. and M.C.; visualization, K.L.-S. and K.G.; supervision, K.L.-S.; project administration, K.L.-S.; funding acquisition, K.L.-S. All authors have read and agreed to the published version of the manuscript.

**Funding:** The work is funded by the Norwegian Financial Mechanism 2014–2021—Small Grant 2020 NOR/SGS//RenMet/0049/2020-00 (11/PE/0146/21), entitled: Innovative hydrometallurgical technologies for the production of rhenium compounds from recycled waste materials for catalysis, electromobility, aviation and defense industry.

**Data Availability Statement:** Not applicable.

**Acknowledgments:** The authors would like to express their thanks for the paid quantitative chemical analyses undertaken in the Łukasiewicz Research Network-Institute of Non-Ferrous Metals, Centre of Analytical Chemistry, and for the XRD qualitative analyses undertaken in the Łukasiewicz Research Network-Institute of Non-Ferrous Metals, Centre of Functional Materials (especially to Lukasz Hawełek). The authors would also like to thank Julita Sztandera for conducting some experiments.

**Conflicts of Interest:** The authors declare no conflict of interest. The funders had no role in the design of the study; in the collection, analyses, or interpretation of data; in the writing of the manuscript; or in the decision to publish the results.

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
