# Peer review of "A New Method of Obtaining High Purity Nickel(II) Perrhenate from Waste"

_crystals, doi:10.3390/cryst13101465_

Round 1

Reviewer 1 Report

This manuscript describes the synthesis of Ni(II) perrhenate from what is more or less waste product. The method is well described and the utility of the compound in preparing alloys that are stable at high temperature is well described.  I find the manuscript clearly written and I have no negative comments apart from minor English improvement.  I recommend publication.

The English is quite good but not perfect.

Author Response

Dear Sir/Madam

Thank you for your kind words and taking the time to read our publication. We would like to inform you that we have carried out a language correction.

Best regards

Authors

Reviewer 2 Report

Manuscript ID: crystals-2614172

Title: A new method of obtaining high purity nickel(II) perrhenate from waste

Authors: Katarzyna Leszczyńska-Sejda et al.

Dear authors, my remarks will be in a very prosaic, businesslike style, however, please do introduce each of them with a "please". I am an advocate of directness in my comments, and I apologize in advance if any of them seem too harsh.

Line 43, 50, 93. Avoid more than 3 references for a fact. A maximum of 3 in a sentence is allowed for Crystals. Describe this information in detail.

Line 171-174. This information can be added to the section 2.

Figure 4. Black and white printed hard copies by a reader will not show colors. So change to solid dotted and dashed lines if possible, keeping the color as well. Authors should add the error bars to the each column.

Authors must use the OriginPro software to provide XRD and diagrams. All figures in the article have low quality and resolution. It is necessary that the figures were at least 300 DPI.

Figure 6. What is the particle size distribution of powders after roasting? Why didn’t authors used SEM for analyze the shape of the particles? The same question about TG/DSC method.

Line 305-318. Add equations for roasting process.

Figure 8. What are the technological parameters of the process? Add this information to the flow sheet.

Section 5. This information can be removing.

References. In manuscript a high proportion of the cited references belong to Authors. Self-citation rate of about 50%. 13 of 27 are from authors. Could you please check whether the inclusion of each of these references is appropriate?

Authors must add more recent articles from 2021-2023 (5-7 links).

Author Response

Dear Sir/Madam,

Please find the full answer in the attachement.

Reviewer 3 Report

A new method of obtaining high purity nickel(II) perrhenate  from waste is very interesting paper. Some minor  improvement are required.

Line 18: anhydrous form, with the following composition: 10.5% of Ni, 18 and 66.6% of Re. Is chemical composition of the the produced high purity nickel(II) perrhenate similar to  the product by MERCK, or one from other companies. Did you compare it?

Line 65: They were described by two groups of scientists (please to complete thise sentence)

Line 96: research was undertaken on the possibility of producing nickel(II) perrhenate entirely from waste. Which type of waste?

Line 105: Ammonium perrhenate of catalytic purity Can you write chemical formula (Ni(ReO4)2 ??

Line 136: a furnace (Producer, Type, Country)

Line 334, 335: The reaction was carried out for 2 hours, at a temperature ranging from 60 to 80°C, until complete dissolution of NiO  and at the same time until the acid was neutralized (pH 5-8). Can you write chemical reaction for formation of nickel (II)-perrhenate

Line 336: at the same time until the acid was neutralized (pH 5-8). What was the neutralisation agent?

Line 358: obtained during Cu production in Poland (Which process: solvent extraction or electrolysis?)

Author Response

Dear Sir/Madam

Thank you for your input and comments. We tried to add as many of them as possible. Below we are sending a short description of all the changes and our comments:

- the composition of our nickel(II) perrhenate depends on its application (catalysis, production of superalloys or the defense industry). It is a high-purity compound, as required by the above-mentioned applications. In addition to Ni and Re, the following impurities have been determined for this product: 66.6% of Re, <5ppm of Bi, As, Zn, Cu; <10ppm Co, Mg, Fe, K, Pb, Na, Ca, Mo (line 19 and 366-368). The Re and Ni content, as well as the color and XRD analysis, clearly confirm its anhydrous form. To the best of our knowledge, MERCK only offers ammonium, sodium, potassium and silver perrhenates (checked on 24/09/2023), so we cannot comment on this part of the review. Our nickel(II) perrhenate is offered by Innovator from Poland;

- the two groups of scientists were added to the line 65;

- the waste used in the process was Ni-containing sulfates semi-finished products of a complicated composition from polish Cu smelter. The full analysis of this material is presented in section 3;

- the chemical formula of ammonium perrhenate was added to line 105;

- the information regarding the furnace has been added to the article (line 138);

- a reaction equation of the formation of nickel(II) perrhenate from perrhenic acid and nickel(II) oxide was added;

- NiO was both the reagent and the neutralizing agent of the process. When the whole perrhenic acid reacted with the NiO, the solution was neutralized to pH 5-8. Information is in the text (line 380-381);

- there is one copper producer in Poland, KGHM Polska Miedź S.A. Copper is recovered mainly from national concentrates. There are 3 Cu smelters, including 2 based on flash furnace technology and 1 with a shaft furnace. In all smelters, copper is recovered using electrorefining. In Poland, processes based on solvent extraction are not used for Cu production.

Thank you again for your positive review contribution to our article.

Best regards

Authors

Round 2

Reviewer 2 Report

The authors are under the misconception that particle shape is not an important aspect of hydrometallurgical research. For example, in alumina production, in addition to chemical content, phase composition, particle size distribution and particle shape are significant aspects. 

In the rest of the questions, the authors have made significant changes, the article can be accepted.